# Commuting to Work: Nucleolar Long Non-Coding RNA Control Ribosome Biogenesis from Near and Far

**DOI:** 10.3390/ncrna7030042

**Published:** 2021-07-14

**Authors:** Victoria Mamontova, Barbara Trifault, Lea Boten, Kaspar Burger

**Affiliations:** 1Mildred Scheel Early Career Center for Cancer Research (Mildred-Scheel-Nachwuchszentrum, MSNZ), University Hospital Würzburg, Josef-Schneider Str. 2, 97080 Würzburg, Germany; victoria.mamontova@uni-wuerzburg.de (V.M.); barbara.trifault@uni-wuerzburg.de (B.T.); lea.boten@stud-mail.uni-wuerzburg.de (L.B.); 2Department of Biochemistry and Molecular Biology, Biocenter of the University of Würzburg, Am Hubland, 97074 Würzburg, Germany

**Keywords:** long non-coding RNA, RNA polymerase II, nucleolus, ribosome biogenesis

## Abstract

Gene expression is an essential process for cellular growth, proliferation, and differentiation. The transcription of protein-coding genes and non-coding loci depends on RNA polymerases. Interestingly, numerous loci encode long non-coding (lnc)RNA transcripts that are transcribed by RNA polymerase II (RNAPII) and fine-tune the RNA metabolism. The nucleolus is a prime example of how different lncRNA species concomitantly regulate gene expression by facilitating the production and processing of ribosomal (r)RNA for ribosome biogenesis. Here, we summarise the current findings on how RNAPII influences nucleolar structure and function. We describe how RNAPII-dependent lncRNA can both promote nucleolar integrity and inhibit ribosomal (r)RNA synthesis by modulating the availability of rRNA synthesis factors in trans. Surprisingly, some lncRNA transcripts can directly originate from nucleolar loci and function in cis. The nucleolar intergenic spacer (IGS), for example, encodes nucleolar transcripts that counteract spurious rRNA synthesis in unperturbed cells. In response to DNA damage, RNAPII-dependent lncRNA originates directly at broken ribosomal (r)DNA loci and is processed into small ncRNA, possibly to modulate DNA repair. Thus, lncRNA-mediated regulation of nucleolar biology occurs by several modes of action and is more direct than anticipated, pointing to an intimate crosstalk of RNA metabolic events.

## 1. Introduction

Cellular growth and development rely on the orderly expression of thousands of genes in a spatiotemporal manner. The human genome contains approximately 19,000 protein-coding genes that are differentially expressed in individual cells and developing tissues [1]. The genome of *C. elegans* and *D. melanogaster*, however, also contain approximately 19,000 and 140,000 protein-coding genes, respectively [2,3]. Thus, comparing the number of protein-coding genes per se does not serve justice to the stark developmental differences between humans, worms, and flies. Indeed, protein-coding genes comprise only about 1.5% of the total RNA synthesis carried out in human cells. In addition, RNAPII transcribes approximately 60,000 non-coding loci, a number that by far exceeds the 1300 or so non-coding loci in *C. elegans* [4,5,6]. Thus, the non-coding genome provides an additional layer of regulation in concert with epigenetics and post-transcriptional modifications, which complements a rather limited number of protein-coding genes in complex organisms.

Here, we review the surprisingly intimate links between RNAPII and ribosome biogenesis in the nucleolus. Ribosome biogenesis is a prime example of cellular homeostasis and requires RNA transcripts from all three RNA polymerases, including RNAPII-dependent nucleolar lncRNA. Mounting evidence suggests that RNAPII-dependent lncRNA are intertwined with the nucleolus and critical for the fine-tuning of rRNA synthesis. The expression of such transcripts is often induced by cellular stress and deregulated in cancer. We discuss how lncRNA modulate the structure and function of the nucleolus both in trans and in cis, thereby mediating a surprisingly direct crosstalk of RNAPII with the nucleolus that may become a future therapeutic target.

## 2. lncRNA Are Critical Regulators of Gene Expression

Indeed, the discovery of regulatory non-coding RNA transcripts was a major advance in molecular biology [7]. The human ENCODE project mapped tens of thousands RNAPII transcription sites that do not encode messenger (m)RNA [8]. It became clear that RNAPII can produce lncRNA from both intergenic regions, introns, and loci that are antisense to protein-coding genes (Figure 1). Such transcripts are >200 nucleotides (nt) long, capped, and often alternatively spliced and non-polyadenylated [9]. Some lncRNA are encoded from distinct transcription units, while others are generated as promoter-associated transcripts and natural antisense transcripts. lncRNA can also arise as 3′UTR-associated transcripts from terminator regions, often overlapping with or interspersed between coding transcripts, or associated with promoters and enhancers [10]. lncRNA are dynamically expressed, often at low level or in a tissue-specific manner. A wealth of literature describes how lncRNA shape the expression of protein-coding genes [11,12,13]. Here, we wish to turn our attention to lncRNA that shape the nucleolus and discuss how RNAPII-dependent lncRNA influence the expression of rDNA genes to regulate ribosome biogenesis.

## 3. The Nucleolus Is a Multifunctional RNA Metabolic Hub

The nucleolus is a membrane-less subcellular compartment and is best known for producing ribosomes [14,15]. Ribosome biogenesis is a complex and highly regulated process that is fundamental for protein synthesis, growth, and proliferation. Human nucleoli form around nucleolar organiser regions (NORs), which are located on the short arms of the five acrocentric chromosomes 13, 14, 15, 21, and 22. Each NOR contains 50–70 rDNA genes, which are 13 kilobases (kb) and interspersed by a 30 kb intergenic spacer (IGS) (Figure 2a). The rDNA genes are arranged as repetitive units and form the rDNA array, which is flanked by proximal distal junction regions and embedded in a shell of perinucleolar heterochromatin [16]. Each rDNA gene encodes a 47S pre-rRNA precursor that is produced by RNAPI transcription and undergoes a cascade of processing steps in the nucleolus and nucleoplasm (Figure 2b). Pre-rRNA processing ultimately generates mature 18S, 5.8S, and 28S rRNA forms that are assembled with 80 ribosomal proteins into pre-ribosomal subunits and exported to the cytoplasm [17]. Ribosome biogenesis involves numerous pre-rRNA folding, modification, and cleavage steps [18]. Making ribosomes requires >150 RNAPII-dependent small nucleolar (sno)RNA, the RNAPIII-dependent 5S rRNA, and >200 ribosomal production and processing factors [19,20,21]. The multifunctional rRNA synthesis factor Nucleolin (NCL), the snoRNA-associated methyltransferase Fibrillarin, and the 32S pre-rRNA processing factor Nucleophosmin (NPM1) are among the best characterised ribosomal assembly factors [22,23,24]. Interestingly, Fibrillarin has an additional ribonuclease activity that regulates rRNA processing independent of snoRNA-guided methylation. The ribonuclease activity is carried by the glycine/arginine-rich (GAR) domain and depends on the interaction of Fibrillarin with nucleolar phosphoinositides [25,26]. On average, a HeLa cell comprises 2–5 nucleoli that measure 1–5 µm in diameter and contain >1300 nucleolar proteins [27,28,29]. In yeast, each rDNA locus produces 10^3^–10^4^ 47S pre-rRNA transcripts per minute and fuels a steady-state level of 10^6^–10^7^ ribosomes [30,31].

The linearity in the steps of ribosome biogenesis is reflected by the nucleolar tripartite structure. Human interphase nucleoli are multi-layered condensates that contain three sub-compartments: the fibrillar center (FC), the dense fibrillar component (DFC), and the granular component (GC) [32,33]. The production and early processing of pre-rRNA occurs at the interface of the FC and the DFC. Late stages of pre-rRNA processing and the assembly of ribosomal subunits occur in the GC [34]. The nucleolar tripartite structure is tightly controlled during the cell cycle and its integrity is sensitive to stress [35]. The interphase nucleolus breaks down at the onset of mitosis, when RNAPI transcription is impaired by the mitotic cyclin-dependent kinase 1 (CDK1). At the exit from mitosis, pre-rRNA synthesis resumes and the nucleolus reassembles its tripartite structure, when CDK1 activity is suppressed and RNAPI transcription is reactivated [36,37,38]. Along the same lines, the nucleolus disintegrates upon specific induction of nucleolar DNA damage or selective impairment of RNAPI transcription [39,40,41,42]. The inhibition of early steps in pre-rRNA processing also leads to nucleolar stress, along with the nucleoplasmic translocation of nucleolar proteins and the formation of ‘nucleolar caps’ [16,43]. Upon nucleolar disintegration, excessive amounts of free ribosomal proteins trigger the impaired ribosome biogenesis checkpoint, stabilise the tumour suppressor p53, and arrest cells in the G_1_-phase [44,45,46,47]. Thus, functional RNAPI transcription and pre-rRNA processing are critical for nucleolar integrity.

Intriguingly, however, in 1974, the Roeder lab noticed that the inhibition of RNAPII with the fungal poison α-Amanitin disrupts nucleoli, although the activity of RNAPI is not affected by the drug [48]. This suggests a role of RNAPII transcription in supporting nucleolar integrity. Indeed, the inhibition of transcriptional kinases like CDK9 and the subsequent depletion of RNAPII-dependent nucleolar snoRNA causes pre-rRNA processing defects that trigger nucleolar disintegration [49,50]. Is it possible that the synthesis of RNAPII-dependent transcripts other than snoRNA contributes to nucleolar integrity?

## 4. lncRNA Modulate Ribosome Biogenesis in Trans

There is increasing evidence that trans-acting lncRNA facilitate a regulatory crosstalk of RNAPII transcription with the nucleolus. The protein-coding gene TBRG4 encodes the intronic, snoRNA-containing lncRNA transcript SLERT, which functions as a critical regulator of ribosome biogenesis [51]. SLERT contains the sequence of the two H/ACA box family snoRNA and accumulates in the nucleolus to promote rRNA synthesis in complex with the nucleolar DEAD-box helicase DDX21 (Figure 2c). DDX21 is a global regulator of the RNA metabolism. The multifunctional enzyme senses the transcriptional status of both RNAPI and RNAPII to coordinate ribosome biogenesis with the expression of protein-coding genes [52]. In the nucleolus, DDX21 forms a ring-shaped structure around RNAPI molecules and impairs pre-rRNA synthesis as an allosteric inhibitor of RNAPI. The binding of SLERT to DDX21 in the nucleolus triggers a conformational change in DDX21, which displaces DDX21 from RNAPI and stimulates transcription of the rDNA locus and pre-rRNA processing [51,53]. In contrast, the depletion of SLERT by antisense oligonucleotides increases the interaction of DDX21 with RNAPI and impairs rRNA synthesis. SLERT is overexpressed in human embryonic stem cells and many cancer cells, reflecting the increased demand for ribosome biogenesis in highly proliferating cells.

Similarly, the lncRNA LETN is essential for nucleolar integrity and sustained proliferation of cancer cells [54] (Figure 2c). LETN is a primate-specific, RNAPII-dependent, and polyadenylated lncRNA that is approximately 4500 nt long and upregulated in embryonic tissues and cancer cells. LETN specifically enriches in the GC of the nucleolus to directly interact with NPM1 and scaffold the oligomerisation of NPM1, which functions as a pentamer [55]. The depletion of LETN phenocopies the knockdown of NPM1 and triggers a nucleolar stress phenotype that includes nuclear disintegration, reduction of rRNA synthesis, loss of NPM1 pentameres, and impaired cell proliferation [54]. The authors further tested the physiological relevance of LETN for ribosome biogenesis and cancer progression. Indeed, patients that suffer from liver cancer have a better prognosis if they express lower levels of LETN compared with patients with high levels of LETN, which correlates with poor prognosis.

Various other trans-acting RNAPII-dependent lncRNA function as structural transcripts to support nucleolar integrity. The McStay lab determined a 207 kb stretch of the DNA sequence immediately proximal and 379 kb distal to the rDNA array, and assembled consensus proximal junction (PJ) and distal junction (DJ) contigs from sequenced bacterial artificial chromosomes [56]. Both PJ and DJ sequences are conserved and present on all five human NOR-containing chromosomes. The authors established the chromatin and transcriptional profile of the DJ consensus sequence and found that the DJ region has a complex chromatin landscape with signatures for promoters, enhancers, and actively transcribed genes. The DJ region is dominated by a large inverted-repeat sequence that is transcribed by RNAPII and encodes for two spliced and polyadenylated lncRNA transcripts called disnor187 and disnor238. Intriguingly, both transcripts localise in the perinucleolar heterochromatin to anchor rDNA repeats and support nucleolar integrity [57]. The depletion of disnor transcripts triggers nucleolar disintegration and diminishes RNAPI activity, suggesting a regulatory role in ribosome biogenesis. However, more studies are required to unravel the precise metabolism of disnor transcripts and their role for nucleolar integrity.

Alu-repeat transcripts are another class of lncRNA that modulate the nucleolus. Alu elements are frequent repetitive elements in the genome of mammals and can be transcribed by both RNAPII and RNAPIII [58,59]. Some Alu-repeat transcripts undergo processing and modulate gene expression at various levels [60]. Interestingly, a subset of intronic Alu-repeat transcripts, which are a by-product of pre-mRNA splicing, accumulate as 100–300 nt double-stranded lncRNA and enrich in the nucleolus, where they interact with both NCL and NPM1 to enhance both RNAPI transcription and pre-rRNA processing [61]. The depletion of intronic Alu-repeat transcripts downregulates RNAPI activity and reduces the size of the nucleolus. These examples illustrate how RNAPII-dependent lncRNA support nucleolar integrity.

However, deregulated activity of RNAPII and an aberrant RNA metabolism can be pathogenic [62]. Several lncRNA species interfere with the nucleolar structure and function. Deep-sequencing of nucleolar RNA in neurons identified the RNAPII-dependent, 1500 nt long lncRNA LoNA [63]. LoNA comprises a structural analogy to SLERT, as it contains an NCL binding site and two C/D box family snoRNA sequences. Unlike SLERT, LoNA negatively regulates ribosome biogenesis (Figure 2c). Nucleolar LoNA binds and inactivates both NCL and the C/D box snoRNA binding factor Fibrillarin, which interferes with transcription, processing, and modification of pre-rRNA. Thus, high levels of LoNA function as a molecular sponge, which inhibits rRNA synthesis at various levels and triggers nucleolar stress. The depletion of LoNA, in turn, enhances ribosome biogenesis, which underscores its inhibitory role for ribosome biogenesis. LoNA levels are increased in a mouse model for Alzheimer’s disease, where the demand for ribosomes is greatly reduced [64]. Conversely, LoNA levels are decreased in the mouse hippocampus, a region of enhanced neural activity that requires the de novo synthesis of ribosomes [63].

Some neurogenerative diseases like dementia and amyotrophic lateral sclerosis are further characterised by an abnormal expansion of a repeated GGGGCC hexamer sequence in the non-coding region of the *C9orf72* gene [65,66]. The expanded hexamer forms stable, four-stranded secondary structures in the DNA, which rely on non-canonical base pairing of the four G bases and are called G-quadruplex (G4), which are resistant to DDX21-mediated resolution [67]. As a result, G4 structures are hard to transcribe and prevent RNAPII from productive elongation [68]. Moreover, the abortive lncRNA transcript derived from the G4-containing *C9orf72* locus also forms G4 structures, which causes the stabilisation of the lncRNA in nucleoplasmic foci. Intriguingly, the authors could show that one of the RNA-binding proteins (RBPs) that associates with the abortive transcript is NCL. The mislocalisation of NCL causes nucleolar stress, impairs ribosome biogenesis, and triggers the toxic accumulation of untranslated mRNA in patients with neurodegenerative diseases.

Circular RNA are RNAPII-dependent lncRNA species that form upon back-splicing of pre-mRNA and regulate various pathophysiological processes in mammals, for instance, by sponging of transcripts or scaffolding of proteins [69,70]. One such transcript, the circular antisense lncRNA circANRIL, accumulates in the nucleolus and sequesters the nucleolar protein Pescadillo1 (Pes1) [71] (Figure 2c). Pes1 is part of the Pes1-Bop1-Wdr12 (PeBoW) complex, which is required for early pre-rRNA processing and the turnover of the 32S pre-rRNA intermediate [72,73,74]. The binding of Pes1 to circANRIL occurs via a 47S pre-rRNA homology domain and impairs the formation of the PeBoW complex. The resulting nucleolar stress is accompanied with smaller nucleolar size and decreased proliferation [71].

In summary, RNAPII-dependent lncRNA are a major determinant of the nucleolar structure and function. Such transcripts can either support the formation of hyperactive nucleoli in fast proliferating cells like cancer, or be pathologic by diminishing nucleolar integrity, as observed in many neurodegenerative diseases. The fine-tuning of nucleolar functions often occurs via scaffolding and sponging function, but it is currently unclear how the interactions with nucleolar proteins are regulated and how widespread the impact on gene expression is.

## 5. lncRNA Modulate Ribosome Biogenesis in Cis

The nucleolus is multi-functional [75]. Besides its stress-sensing feature that triggers the impaired the ribosome biogenesis checkpoint, the nucleolus also functions as homeostatic buffer. Upon heat shock, for instance, numerous non-nucleolar proteins associate with the pre-rRNA processing factor NPM1 to accumulate in the nucleolus [76]. The non-disintegrated nucleolus serves as a protein quality control compartment that stores misfolded proteins by a chaperone-like capacity to mitigate cellular stress. Intriguingly, many of the 200 or so proteins that interact with NPM1 comprise features of RBPs. It is tempting to speculate that lncRNA other than rRNA may, at least in part, support the homeostatic activity of the nucleolus.

### 5.1. RNAPI-Dependent Nucleolar lncRNA

Indeed, nucleolar sequestration of RBPs has been linked to the accumulation of nucleolar lncRNA. Such transcripts are produced by RNAPI upon various kinds of stress, including heat shock. Surprisingly, however, they are not encoded in the rDNA locus, but arise from specific regions in the IGS that are transcribed in a stress-specific manner. IGS-derived lncRNA sequesters a subset of nucleoplasmic proteins, such as the E3-ligase HDM2, in the nucleolus, which triggers a cellular stress response that involves the stabilisation of downstream effectors like p53 [77,78]. Further, the IGS encodes the >10 kb RNAPI-dependent lncRNA PNCTR, which contains multiple binding sites for the pre-mRNA processing factor PTBP1 [79]. The binding of PTBP1 to PNCTR sequesters PTBP1 in a distinct body in close proximity to the nucleolus called perinucleolar compartment (PNC). The expression of PNCTR and the formation of the PNC body is enhanced by oncogenic stress in cancer cells and promotes pre-mRNA splicing events that favour elevated growth and proliferation. Thus, the homeostatic function of the nucleolus may, at least in part, depend on the production of non-ribosomal IGS transcripts by RNAPI.

The stress-induced production of IGS transcripts is an intriguing addition to the well-described rDNA silencing function of promoter-associated RNA (pRNA) in unperturbed cells, where only about 50% of rDNA genes comprise open chromatin and are transcribed by RNAPI [80]. pRNA are 150–350 nt long, RNAPI-dependent ncRNA transcripts that are produced from an alternative promoter in the IGS region upstream of the rDNA promoter. pRNA form RNA-DNA triple-helix structures at the rDNA promoter and are part of the nucleolar remodelling complex (NoRC) that facilitates silencing of rDNA loci by methylation of histones and DNA to maintain rDNA stability [81,82,83]. The formation of heterochromatic rDNA repeats further requires several ribonucleoprotein complexes that include enzymes to modify both histones (e.g., the histone deacetylase Sir2 and the histone methyltransferase Set1 in *S. cerevisiae*) and DNA at CG-rich regions at the rDNA promoter (e.g., the DNA methyltransferases DNMT1/DNMT3b in humans) [84,85,86,87,88].

Interestingly, the silencing of yeast 35S rDNA loci prevents both spurious RNAPII activity and unwanted recombination events at nucleolar loci [89,90]. RNAPI and RNAPII compete for access to nucleolar chromatin in *S. cerevisiae* [91,92] and the 35S rDNA locus can even undergo an RNA polymerase switch in the synthesis of rRNA, where yeast RNAPII transcribes the 35S rDNA gene to promote survival upon mutation of yeast RNAPI subunits [93,94]. Likewise, the depletion of Sir2 impairs the formation of heterochromatin in the yeast non-transcribed spacer (NTS), which is accompanied by increased occupancy of yeast RNAPII at endogenous transcription units in the NTS and accumulation of NTS-derived lncRNA [95,96].

### 5.2. RNAPII-Dependent Nucleolar lncRNA

Could the human nucleolus also give raise to lncRNA that are produced by RNAPII directly from nucleolar loci? And if so, what are their functions? Indeed, the loss with DNA methylation at the rDNA locus enhances RNAPII transcription in human cells [97]. The co-deletion of DNMT1 and DNMT3b causes a loss of CpG methylation at the rDNA promoter and a reduction of RNAPI occupancy and activity within the 47S rDNA transcription unit. In turn, RNAPII occupancy strongly increases across the rDNA locus. The reduction in pre-rRNA synthesis increases the production of sense-oriented, RNAPII-dependent lncRNA from the rDNA locus, which partially disrupts ribosome biogenesis and causes nucleolar stress. Thus, RNAPII can transcribe sense-oriented lncRNA from nucleolar loci.

Moreover, the mammalian IGS also encodes RNAPII-dependent lncRNA antisense transcripts. The promoter and pre-rRNA antisense transcripts (PAPAS) are well characterised. PAPAS are a heterogenous population of 12–16 kb lncRNA that originate from random antisense promoters in the 5′ region of the IGS [98] (Figure 3a). PAPAS are transcribed by RNAPII upon various kinds of stimuli, such as heat shock, or serum starvation, and form a complex with chromatin remodelling factors and histone deacetylase to heterochromatinise the rDNA locus and downregulate rRNA synthesis [99]. The silencing effect of PAPAS transcripts depends on the formation of an RNA-DNA triple-helix structure and involves the nucleosome remodelling and deacetylation (NuRD) complex [100].

More recently, a direct function for nucleolar RNAPII transcription in promoting rRNA synthesis and nucleolar integrity has been reported [101]. The authors combine high-resolution imaging and chromatin immunoprecipitation with next-generation sequencing and genome editing to map the localisation and activity of nucleolar RNAPII at the rDNA locus of human cells (Figure 3b). Surprisingly, nucleolar RNAPII occupancy was detected across the rDNA locus and specifically enriched in the IGS region. Nucleolar RNAPII is active in unperturbed cells and transcribes antisense lncRNA from the IGS region. The RNAPII-dependent transcripts hybridise with nucleolar IGS DNA in cis to form an RNA-DNA hybrid (or R-loop). The formation of R-loops across the IGS region is promoted by the neurodegeneration-associated protein Senetaxin and represents a stable DNA-RNA hybrid structure that blocks the recruitment of RNAPI to the IGS region. This prevents the synthesis of RNAPI-dependent sense intergenic non-coding RNA (sincRNA). The authors could further show that the interference with RNAPII activity by chemical inhibitors or the suppression of R-loops by overexpression of RNaseH1 stabilises sincRNA. The accumulation of sincRNA triggers nucleolar stress, as observed by disintegration of nucleoli and translocation of nucleolar proteins. Importantly, the levels of sincRNA are elevated in an Ewing sarcoma (EWS) cancer model system, where mutations in the EWS RNA-binding protein 1 (EWSR1) perturb the metabolism of nucleolar R-loops. Indeed, rRNA synthesis is defective and the nucleolar structure appears disorganised in EWS patient samples. Strikingly, the defects could be rescued by the depletion of sincRNA. These findings suggest a mechanism where RNAPII competes with RNAPI for transcription of nucleolar loci, with RNAPII-dependent antisense transcripts forming an ‘R-loop shield’ to prevent spurious RNAPI activity and foster efficient ribosome biogenesis.

## 6. DNA Damage Stimulates Nucleolar RNAPII Transcription

The loss of CpG methylation upon co-deletion of DNMT1 and DNMT3a strongly increases the recombination rate of human rDNA loci, but does not induce DNA damage [97], suggesting that nucleolar lncRNA may be produced by RNAPII to promote genome stability. The functional relevance of RNAPII-dependent lncRNA synthesis for the DNA damage response has recently been tested by locus-spec induction of DNA double-strand breaks (DSBs). The endonuclease AsiSI, for example, preferentially cleaves in close proximity to RNAPII promoters. AsiSI cleavage triggers a lncRNA-dependent DNA damage response that involves the production of damage-induced (di)lncRNA by RNAPII directly at promoter-associated DSBs. dilncRNA is >1000 nt long and undergoes processing into double-stranded, damage-induced small RNA (diRNA or DDRNA) via RNA interference (RNAi) factors Drosha and Dicer in the nucleus [102,103,104,105,106]. The Dicer-dependent accumulation of diRNA/DDRNA is required for the efficient recruitment of a subset of DNA repair factors to DSBs in mammals [107,108]. The endonuclease I-PpoI cleaves the rDNA gene within the 28S rDNA region and induces prominent nucleolar DSBs. Strikingly, RNAPII can produce 1000–2500 nt lncRNA directly from the antisense strand of broken 28S rDNA loci (Figure 3c). These transcripts are also processed into diRNA/DDRNA in a partially Dicer-dependent manner [109]. The damage-induced production and processing of non-ribosomal nucleolar transcripts has also been reported in *N. crassa* [110]. The treatment with DNA methylating agents like ethyl methanesulphonate or methyl methanesulphonate results in the accumulation of aberrant lncRNA (aRNA) that originate from the entire rDNA locus and, again, undergo processing into small-interfering RNA by components of the fungal RNAi machinery. The authors could show that RNAi mutants interfere with aRNA processing and render *N. crassa* hypersensitive to DNA damage. These findings point towards an RNA-dependent crosstalk of the nucleolus with the DNA damage response. However, future studies are required to further characterise the relevance of nucleolus-derived transcripts for gene expression.

## 7. Concluding Remarks

In 1896, Pianese identified enlarged nucleoli as a feature of cancer cells [111]. Ribosome biogenesis is controlled by various major proto-oncogenes and tumour suppressors, such as c-Myc, p53, and the retinoblastoma protein Rb, which deregulate ribosome biogenesis to meet the increased demand of protein synthesis in cancer cells [112,113]. The proto-oncogene c-Myc, for example, amplifies both the production and the processing of pre-rRNA. More than 60 c-Myc target genes belong to the class of ribosomal synthesis and assembly factors [114]. c-Myc also directly stimulates the rate of RNAPI transcription at the rDNA promoter [115]. On the other hand, impaired ribosome biogenesis is pathogenic. Loss of function mutations of various nucleolar transcription, pre-rRNA processing factors, and ribosomal proteins (e.g., Treacle/TCOF1, Dyskerin, RPS14) causes a variety of syndromes collectively termed as ‘ribosomopathies’. Diseases like Teacher Collins syndrome are characterised by dysfunctional ribosomes and defects in the fidelity of translation, which trigger haematological defects and cancer predisposition [116]. Thus, there is increasing biomedical interest in the nucleolus, which could be considered as the ‘Achilles heel’ of tumourigenesis. Could the regulation of nucleolar integrity by RNAPII-dependent lncRNA be exploited biomedically? A mounting body of evidence suggests that RNAPII-dependent lncRNA shapes the nucleolus at different scales, in particular in response to cellular stress. The number of non-ribosomal lncRNA transcripts that modulate ribosome biogenesis is steadily increasing [117,118,119]. Firstly, a flurry of trans-acting lncRNA, which are produced in the nucleoplasm and accumulate in the nucleolus, specifically interact with nucleolar proteins to modulate ribosome biogenesis. Secondly, some trans-acting lncRNA species can act as a ‘molecular glue’ to scaffold nucleolar structure and promote phase separation. Thirdly, RNAPII is directly engaged in the nucleolus to fine-tune rRNA synthesis in cis. We have discussed examples where RNAPII-dependent lncRNA can both stimulate the formation of hyperactive nucleoli (e.g., SLERT or Alu-repeat transcripts in cancers) and cause the disintegration of nucleoli by impairing the functions of nucleolar proteins (e.g., LoNA or the C9orf72 transcripts in neurodegenerative diseases). The specific depletion of these lncRNA by antisense oligonucleotides may be a promising approach for the non-genotoxic interference with nucleolar biology and gene expression. Moreover, the selective RNAPI inhibitor CX-5461 triggers a potent nucleolar stress response and recently advanced in clinical trials [120]. It will be important to investigate whether the combination of RNAPI inhibitors with antisense oligonucleotide technology that interferes with nucleolar lncRNA may potentiate the nucleolar stress response and pave the way for novel therapeutic strategies.

Nevertheless, we have just begun to understand the relevance of RNAPII for modulating the expression of rDNA genes and the integrity of the nucleolus. What are the regulatory principles that control the activity of nucleolar RNAPII and the fate of nucleolar RNAPII-dependent lncRNA? What enzymes regulate RNAPII activity in the nucleolus? How does RNAPII find a nucleolar promoter? Is the regulatory function of such transcripts due to a quantitative change in level or are there also qualitative differences? Recent findings in the DNA repair field suggest that RNAPII can recognise DSBs irrespective of the genomic context by the assembly of a de novo RNAPII promoter that contains the TATA-binding protein TBP as well as the transcriptional kinase CDK9 [104,121]. Interestingly, a 55 kDa large isoform of CDK9 (CDK9-55k) originates from an alternative upstream promoter and localises throughout the nucleus [122,123]. CDK9-55k interacts with the DNA-end binding protein Ku70 and the depletion of CDK9-55k stabilises DSBs [124]. Further, DNA-PK, one of the three major DSB-sensing kinases, is a known activator of RNAPII transcription and enriches in the nucleolus to mediate pre-rRNA processing [125,126,127]. It is tempting to speculate that the localisation and activity of nucleolar RNAPII may be altered in tumours and regulated by damage-responsive kinases [128]. To further understand the surprisingly direct functions of RNAPII in the nucleolus, it will be relevant to biochemically characterise the nucleolar RNAPII complex and deduce potential differences to RNAPII engaged on protein-coding genes. In sum, the investigation of nucleolar biology remains an exciting prospect that may hold more surprises at the crossroads of RNA biology.

## Figures and Tables

**Figure 1 ncrna-07-00042-f001:**
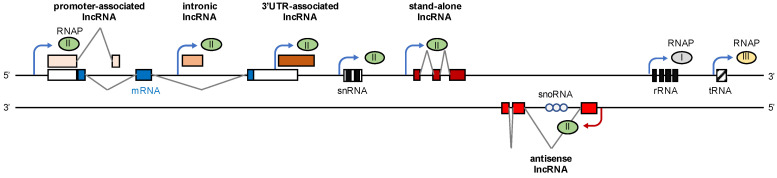
Transcription units of the genome. The human genome engages three RNA polymerases (RNAPs) for RNA synthesis. RNAPII (green) produces both protein-coding messenger (m)RNA transcripts and long non-coding (lnc)RNA transcripts from sense (blue arrowhead) and antisense (red arrowhead) promoters. RNAPII also produces small nuclear (sn)RNA and small nucleolar (sno)RNA. RNAPI (grey) and RNAPIII (yellow) produce non-coding ribosomal (r)RNA and transfer (t)RNA, respectively. White box, non-protein-coding exon; blue box, protein-coding exon; red box, lncRNA-coding exon; 3′UTR, 3′ untranslated region. The transcription units are not in scale.

**Figure 2 ncrna-07-00042-f002:**
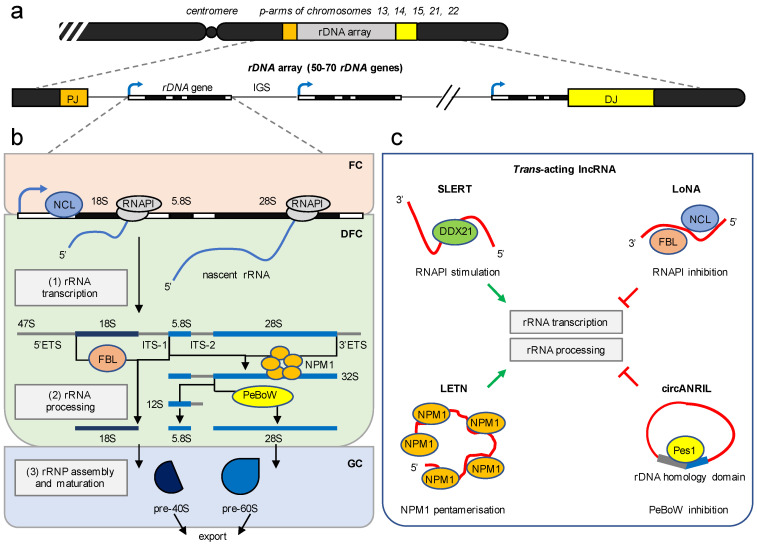
Structural and functional organisation of ribosome biogenesis by lncRNA. (**a**) Schematic of the rDNA array on the p-arms of five acrocentric chromosomes. PJ, proximal junction; DJ, distal junction; blue arrowhead, rDNA promoter; IGS, intergenic spacer. The transcription units are not to scale. (**b**) Ribosome biogenesis requires three major steps and occurs in the context of the nucleolar tripartite structure. (1) rRNA transcription requires synthesis of an rRNA precursor by RNAPI and occurs at the interface of the fibrillar center (FC) and the dense fibrillar component (DFC). (2) rRNA processing occurs in the DFC and involves a cascade of cleavage and trimming steps that remove ribosomal spacer sequences and produce mature 18S, 5.8Sm and 28S rRNA forms via 32S and 12S rRNA intermediates. 5′ETS/3′ETS, 5′/3′ external transcribed spacer; ITS-1/ITS-2, internal transcribed spacer-1/-2. NCL, Nucleolin; FBL, Fibrillarin; NPM1, Nucleophosmin 1; PeBoW, Pes1-Bop1-Wdr12 complex are major ribosomal synthesis factors. (3) The assembly and maturation of ribosomal ribonucleoprotein complexes (rRNPs) occur in the granular component (GC) and generate 40S and 60S ribosomal subunit precursors (pre-40S, pre-60S). (**c**) Examples of trans-acting nucleolar lncRNA. SLERT and LETN stimulate (green arrowhead) rRNA synthesis by exclusion of the nucleolar DEAD-box helicase 21 (DDX21) from RNAPI and scaffolding the pentamerisation of NPM1, respectively. LoNA and circANRIL inhibit (red block) rRNA synthesis by sequestration of FBL, NCL, and Pes1, respectively. Refer to the main text for details.

**Figure 3 ncrna-07-00042-f003:**
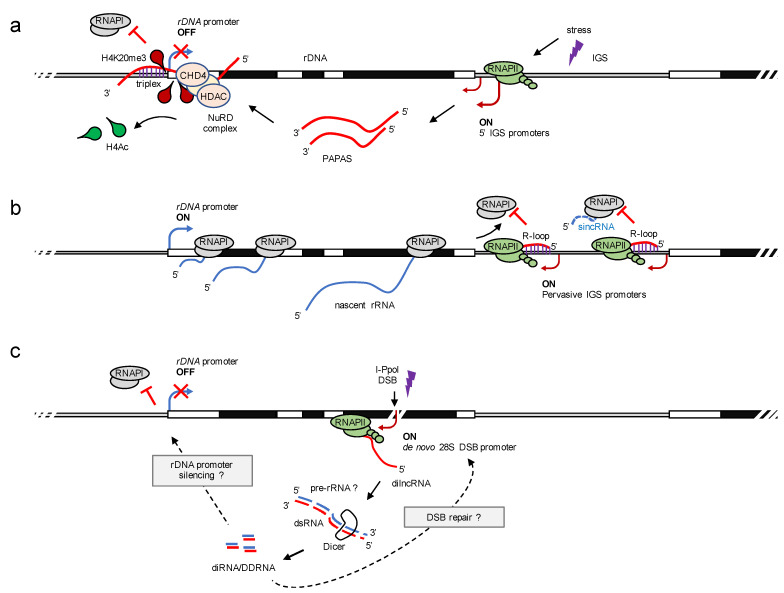
Regulation of rRNA synthesis by cis-acting lncRNA. (**a**) lncRNA-mediated downregulation of rRNA synthesis. Cellular stress activates transcription of promoter and pre-rRNA antisense transcripts (PAPAS, red) from random antisense RNAPII promoters (red arrowhead) in the 5′ region of the IGS. PAPAS are tethered to the rDNA enhancer via triple-helix formation and interact with chromatin remodelling factors such as the chromodomain-helicase-DNA-binding protein 4 (CHD4) and histone deacetylases (HDAC) to form the nucleosome remodelling and deacetylation (NuRD) complex. NuRD represses rRNA synthesis via formation of histone H4 lysine-20 trimethylation (H4K20me3) and removal of histone H4 acetylation (H4Ac) marks at the rDNA promoter (blue arrowhead). (**b**) lncRNA-mediated surveillance of rRNA synthesis. Pervasive transcription of RNAPII across the IGS produces antisense lncRNA (red) that form RNA-DNA hybrids (R-loops). The formation of nucleolar R-loops prevents transcription of the IGS by RNAPI. However, the disruption of nucleolar R-loops enables the recruitment of RNAPI to the IGS and results in the synthesis of sense intergenic non-coding RNA (sincRNA; dashed blue) that mimic nucleolar stress. The suppression of sincRNA transcription maintains the bona fide synthesis of nascent rRNA (blue). (**c**) Nucleolar lncRNA in the DNA damage response. The locus-specific induction of a DNA double-strand break (DSB) in the 28S rDNA by the endonuclease I-PpoI triggers the synthesis of damage-induced lncRNA (dilncRNA) by RNAPII from a de novo antisense promoter (red arrowhead). A subset of dilncRNA forms double-stranded (ds)RNA and is processed by the endonuclease Dicer into small damage-induced RNA (diRNA or DDRNA). The functional relevance of nucleolar diRNA/DDRNA is currently unclear, but may include the recognition and repair of the 28S rDNA DSB, silencing of the rDNA promoter (blue arrowhead), and/or downregulation of RNAPI transcription (red block) in a small RNA-dependent manner (?). The transcription units are not in scale.

## Data Availability

Not applicable.

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
