# Peer review of "Commuting to Work: Nucleolar Long Non-Coding RNA Control Ribosome Biogenesis from Near and Far"

_ncrna, 2021, doi:10.3390/ncrna7030042_

Round 1
Reviewer 1 Report
In this well-written review, the authors describe current ideas linking RNAPII and ribosome biogenesis. The topic is very timely due to its relevance for a better understanding of cellular stress and cancer. The writing style is clear and informative.
Minor suggestions/improvements:
Page 2: the background covered in lines 52-65 is too general and should be reduced - this is after all a specialized review, not a molecular biology undergraduate textbook. The paragraph could easily start with the sentence in line 66, especially as Fig. 1 provides a good graphical summary of the general concepts.
page 7: Line 244: “In summary” is a more conventional way instead of “In sum”
page 8: line 296. This is merely a further suggestion, but would it not be better to start a new section with its own title here?
page 9: line 383/383: “N. crassa” should be written in italics as a species name
page 11: line 410: Would it be a good idea to talk about phase separation here? Presumably, that is how the RNA could act as “molecular glue”.
Author Response
Response to Reviewer #1
In this well-written review, the authors describe current ideas linking RNAPII and ribosome biogenesis. The topic is very timely due to its relevance for a better understanding of cellular stress and cancer. The writing style is clear and informative.
We thank reviewer #1 for the overall very positive comments.
Minor suggestions/improvements:
Page 2: the background covered in lines 52-65 is too general and should be reduced - this is after all a specialized review, not a molecular biology undergraduate textbook. The paragraph could easily start with the sentence in line 66, especially as Fig. 1 provides a good graphical summary of the general concepts.
We agree with the reviewer and have deleted the paragraph (lines 52-65).
page 7: Line 244: “In summary” is a more conventional way instead of “In sum”
We agree with the reviewer and have changed the wording accordingly.
page 8: line 296. This is merely a further suggestion, but would it not be better to start a new section with its own title here?
We appreciate the reviewer’s suggestion to start a new section here. To more clearly distinguish findings that discuss Pol I-dependent lncRNA from Pol II-dependent nucleolar lncRNA functions, we have divided section 5 into two subsections with the following headers: ‘5.1. RNAPI-dependent nucleolar lncRNA’ starting at line 260 and ‘5.2. RNAPII-dependent nucleolar lncRNA’ starting at line 296.
page 9: line 383/383: “N. crassa” should be written in italics as a species name
We agree with the reviewer and have changed the format accordingly.
page 11: line 410: Would it be a good idea to talk about phase separation here? Presumably, that is how the RNA could act as “molecular glue”.
We appreciate the reviewer’s suggestion to talk about phase separation and added the text ‘...to scaffold nucleolar structure and promote phase separation’
Reviewer 2 Report
The submitted manuscript nicely summarizes the current knowledge about the connection between RNAPII-expressed lncRNA and nucleolar structure and function. The topic of the manuscript is of interest for broad audience of ncRNA Journal. However, in the context of nucleolar organization it could be important to discuss nuclear phosphoinositides namely PI4,5P2. This phospholipid is important for RNAPI and RNAPII activity and nucleolar protein fibrillarin localization. For this reason, the authors should take into account following papers and discuss the major findings in short paragraph:
doi.org/10.3390/cells9051143
doi.org/10.3390/cells10010068
doi.org/10.1016/j.bbalip.2021.158890
doi.org/10.1016/j.mex.2021.101372
Author Response
Response to Reviewer #2
The submitted manuscript nicely summarizes the current knowledge about the connection between RNAPII-expressed lncRNA and nucleolar structure and function. The topic of the manuscript is of interest for broad audience of ncRNA Journal.
We thank reviewer #2 for the overall very positive comments.
However, in the context of nucleolar organization it could be important to discuss nuclear phosphoinositides namely PI4,5P2. This phospholipid is important for RNAPI and RNAPII activity and nucleolar protein fibrillarin localization. For this reason, the authors should take into account following papers and discuss the major findings in short paragraph:
doi.org/10.3390/cells9051143
doi.org/10.3390/cells10010068
doi.org/10.1016/j.bbalip.2021.158890
doi.org/10.1016/j.mex.2021.101372
We appreciate the reviewer’s suggestion to discuss phosphoinositides and their roles for Fibrillarin function and transcriptional activity. We have added the following text and citations in line 106: ‘…assembly factors [Ref 26-28]. Interestingly, Fibrillarin has an additional ribonuclease activity that regulates rRNA processing independent of snoRNA-guided methylation. The ribonuclease activity is carried by the glycine/arginine-rich (GAR) domain and depends on the interaction of Fibrillarin with nucleolar phosphoinositides (Guillen-Chable et al., 2020 Cells; Sztacho et al., 2021 Cells). On average…‘
We would like to suggest, however, not to include the two other references from the more technical papers, as we feel that this may be beyond the scope of this review.
Reviewer 3 Report
The manuscript is divided in 7 sections, including an introduction and a concluding remarks section. Section 2 is dedicated to lncRNA that are critical regulators of gene expression. In section 3, the authors summarize the current knowledge on the nucleolus and its role as a RNA metabolic hub. Then the authors focus on ribosome biogenesis and the implication of both trans and cis-acting lncRNA in section 4 and 5 respectively. The last section is dedicated to the impact of DNA damage on nucleolar transcription by RNA polymerase II. The manuscript is illustrated by three informative figures and 129 references. Overall, the manuscript is very well written, the review will definitely be of general interest for the scientific community. Therefore, this reviewer can warmly recommend this manuscript for publication.
Minor points:
- page 6, last paragraph: the authors present the G-quadruplex involved in ALS. A critical reference on the implication of DDX21 is missing in this paragraph. PMID: 28472472 should be included in this paragraph.
- reference format of references 3, 4, 11, 12, 30, 67, 68 is not appropriate.
- page 7, line 248, this sentence requires revision. ‘Its’ replaced by ‘It is’ ?
- fig 2C is a bit minimalist and should if possible include more mechanistic details.
Author Response
Response to Reviewer #3
The manuscript is divided in 7 sections, including an introduction and a concluding remarks section. Section 2 is dedicated to lncRNA that are critical regulators of gene expression. In section 3, the authors summarize the current knowledge on the nucleolus and its role as a RNA metabolic hub. Then the authors focus on ribosome biogenesis and the implication of both trans and cis-acting lncRNA in section 4 and 5 respectively. The last section is dedicated to the impact of DNA damage on nucleolar transcription by RNA polymerase II. The manuscript is illustrated by three informative figures and 129 references. Overall, the manuscript is very well written, the review will definitely be of general interest for the scientific community. Therefore, this reviewer can warmly recommend this manuscript for publication.
We thank reviewer #3 for the overall very positive comments.
Minor points:
- page 6, last paragraph: the authors present the G-quadruplex involved in ALS. A critical reference on the implication of DDX21 is missing in this paragraph. PMID: 28472472 should be included in this paragraph.
We agree with the reviewer and have added text and references in line 226: ‘...called G-quadruplex (G4), which are resistant to DDX21-mediated resolution (McRae et bal., 2017 NAR). As a result, G4…’
- reference format of references 3, 4, 11, 12, 30, 67, 68 is not appropriate.
We appreciate the reviewer’s comment. However, the Zotero reference manger used for this review suggests a full list of authors when formatting in MDPI style. Similarly, the instruction for authors suggest: ‘Note: If you are not sure how to abbreviate a particular journal title, please leave the entire title. The Editorial Office will abbreviate those journal titles appropriately.‘ Thus, we wish to keep the full list of authors for now and leave abbreviations to the editorial office.
- page 7, line 248, this sentence requires revision. ‘Its’ replaced by ‘It is’ ?
We agree with the reviewer and have changed the text accordingly.
- fig 2C is a bit minimalist and should if possible include more mechanistic details.
We appreciate the reviewer’s comment. However, we wish to keep a rather simple, schematic illustration of nucleolar lncRNA functions, as a more detailed illustration in Fig 2C may overload the overall figure 2.
Nevertheless, we have added text in the legend: ‘Refer to main text for details.’